# Prevalence of Fetal Alcohol Spectrum Disorders (FASD) among Children Adopted from Eastern European Countries: Russia and Ukraine

**DOI:** 10.3390/ijerph18041388

**Published:** 2021-02-03

**Authors:** Joan Colom, Lidia Segura-García, Adriana Bastons-Compta, Marta Astals, Vicente Andreu-Fernandez, Natalia Barcons, Raquel Vidal, Ana I. Ibar, Vicky Fumadó, Nuria Gómez, Agnés Russiñol, Oscar Garcia-Algar

**Affiliations:** 1Program on Substance Abuse, Public Health Agency of Catalonia, Department of Health, Generalitat de Catalunya, 08005 Barcelona, Spain; lidia.segura@gencat.cat (L.S.-G.); aibarf@gencat.cat (A.I.I.); 2Maternal and Child Health and Development Network (Red SAMID), Instituto de Salud Carlos III (ISCIII), 28029 Madrid, Spain; adrianabastons@gmail.com (A.B.-C.); marta.astals@gmail.com (M.A.); viandreu@clinic.cat (V.A.-F.); ogarciaa@clinic.cat (O.G.-A.); 3Neonatology Unit, ICGON, IDIBAPS, Hospital Clínic-Maternitat, BCNatal, 08028 Barcelona, Spain; 4Departament de Cirurgia i Especialitats Mèdico-Quirúrgiques, University of Barcelona, 08036 Barcelona, Spain; 5Departament of Nutrition and Health, Valencian International University (VIU), 46002 Valencia, Spain; 6Pediatrics Service, Hospital Sant Joan de Déu, 08950 Barcelona, Spain; nataliabarcons@gmail.com (N.B.); vfumado@sjdhospitalbarcelona.org (V.F.); 7Psychiatry Service, Hospital Universitari Vall d’Hebron, Group of Psychiatry, Mental Health and Addiction, Vall d’Hebron Research Institute (VHIR), 08035 Barcelona, Spain; rvidal@vhebron.net (R.V.); nurgomez@vhebron.net (N.G.); 8Biomedical Network Research Centre on Mental Health (CIBERSAM), 28029 Madrid, Spain; 9Department of Psychiatry and Legal Medicine, Universitat Autònoma de Barcelona, 08193 Barcelona, Spain; 10Department of Labour, Social Affairs and Families, Catalan Institute for Fostering and Adoption, 08001 Barcelona, Spain; agnes.russinol@gencat.cat

**Keywords:** fetal alcohol spectrum disorders (FASD), adopted children, cognitive disorder, neurodevelopment impairment, prenatal exposure to alcohol (PEA), alcohol-exposed pregnancies, alcohol

## Abstract

Fetal alcohol spectrum disorder (FASD) is a leading cause of neurodevelopmental disorders. Children adopted internationally from countries where alcohol consumption during pregnancy is very high are at greater risk for FASD. Lack of expertise in diagnosing FASD and mixed neurodevelopmental and behavioral signs due to abandonment complicate a timely diagnosis. The aim of this study was to determine the prevalence of FASD in adopted children. Children between the ages of 8 and 24 adopted from Russia and Ukraine were evaluated for clinical and historical features of FASD. Of the 162 children evaluated, 81 (50%) met FASD diagnostic criteria. Thirty-three (20.4%) children had fetal alcohol syndrome (FAS), 28 (17.2%) had partial FAS, 2 (1.2%) had alcohol-related birth defects (ARBD) and 18 (11.1%) had alcohol-related neurodevelopmental disorder (ARND). Of the 81 children in which fetal alcohol exposure could not be confirmed, many had manifestations that would have established a diagnosis of FASD if a history of maternal alcohol consumption was confirmed. In a population of children with a high risk of prenatal alcohol exposure (adoptees from Eastern European countries), at least 50% showed manifestations associated with FASD. The reported prevalence in this study is in line with the results obtained in a previous study as well as in orphanages of origin.

## 1. Introduction

### 1.1. Alcohol Consumption and Impact on Health and Harm to Others

Alcohol is a psychoactive substance that has been consumed for centuries in different cultures. Among the population aged 15 to 49 years, alcohol use was the leading risk factor globally in 2016, with 3.8% (95% uncertainty interval, UI 3.2–4.3) of female deaths and 12.2% (10.8–13.6) of male deaths attributable to alcohol use. For the population aged 15 to 49 years, female attributable disability-adjusted life years (DALYs) were 2.3% (95% UI 2.0–2.6) and male attributable DALYs were 8.9% (7.8–9.9) [1].

Alcohol is associated with 230 diseases or conditions of loss of health for the consumer and their social environment, promoting a heavy social and economic burden on society. The harmful effects of alcohol on different populations and communities are determined by the volume of alcohol consumed, habits and frequency of consumption, the quality of alcohol and the genetic background related to its metabolism [2]. The Global Status report [3] establishes that the highest per capita consumption of alcohol (10 liters or more by year) is observed in countries of the WHO European Region. Heavy episodic drinking among alcohol consumers is very high (≥60% of current drinkers) in Russia, Ukraine and the other old republics of the Soviet Union as well as in Bulgaria, Poland and Romania.

### 1.2. Epidemiological Data of Prenatal Exposure to Alcohol (PEA) and FASD

Globally, women consume alcohol in lower amounts and less frequently than men. However, the gender gap has been decreasing and even disappearing in some EU countries [3]. Prenatal alcohol exposure is considered one of the major public health challenges [4]. Alcohol is the most common teratogenic agent in all cultures throughout history. It is considered, excluding the causes of genetic origin, the most important determinant for mental and behavioral disorders. In fact, it is considered the leading preventable, non-genetic cause of mental retardation in the Western world. The deleterious outcomes caused by prenatal exposure to alcohol are related to several different variables such as dose, time, duration and pattern of substance consumption during the different stages of pregnancy, as demonstrated in animal studies [2,5].

Alcohol consumption during pregnancy may result in a series of adverse effects to the fetus including congenital anomalies and behavioral, cognitive and adaptive deficits grouped under the term of fetal alcohol spectrum disorders (FASD). The FASD continuum includes four different disorders: fetal alcohol syndrome (FAS), partial fetal alcohol syndrome (PFAS), alcohol-related neurodevelopmental disorder (ARND) and alcohol-related birth defects (ARBD) (Table 1) [2,6,7,8]. The only sound recommendation related to alcohol consumption during pregnancy is that no amount of alcohol can be considered safe during pregnancy based on research evidence [2].

A global recent systematic review and meta-analysis estimated that the global prevalence of alcohol use during pregnancy in the general population amounts to 9.8% [9]. However, this percentage is usually based on questionnaires, reaching 45% when assessed by biological biomarkers as neonatal meconium or maternal hair [10,11]. In addition, at the country level, it was observed that binge drinking during pregnancy ranged from 0.2% to 13.9% [12]. The European area leads the ranking for alcohol use during pregnancy with a prevalence of 25.2% [3,4,9,12]. Spain ranked third among European countries with a higher prevalence of alcohol consumption during pregnancy (45.0% population) [10,13]. Furthermore, FASD prevalence could still be underestimated because of numerous undiagnosed and misdiagnosed cases.

The last published systematic review about the prevalence of alcohol consumption during pregnancy and fetal alcohol syndrome (FAS, defined by growth retardation, facial malformations and central nervous system impairment) indicated that the five countries with the highest estimated prevalence of alcohol use during pregnancy were Ireland (60.4%), Belarus (46.6%), Denmark (45.8%), the UK (41.3%) and Russia (36.5%) [9,12].

In relation to the prevalence of FAS, the systematic review by Popova indicates that the five countries with the highest prevalence of FAS are South Africa (58.5 per 1000), Croatia (11.5 per 1000), Ireland (8.9 per 1000), Italy (8.0 per 1000) and Belarus (6.9 per 1000) [3,4,9,12]. In a previous study in Italy, FASD prevalence in school children was found to be as high as 4.0–7.0% [14]. In two recent prevalence screening studies in the UK, prevalence of FASD was found to be 17% in the general population and 25% in adopted children [15,16].

The global prevalence of FAS among the general population is estimated to be 1.46 per 1000 (95% CI 9.4–23.3) [4,9]. One in every 67 women who consumed alcohol during pregnancy would deliver a child with FAS, which translates into about 119,000 children born with FAS in the world every year [9]. Moreover, the global prevalence of FASD among children and youth was 7.7 per 1000 in the general population (95% CI, 4.9–11.7 per 1000 population), with the European region having the highest overall prevalence at 19.8 per 1000 population (95% CI, 14.1–28.0 per 1000 population) [12].

Another recent systematic review and meta-analysis found that 428 conditions (which spanned 18 of the 22 International Classification of Diseases (ICD)-10 chapters) co-occurred with FASD [17], some of the most common health problems being congenital malformations, chromosomal abnormalities, prenatal and postnatal growth delays, intellectual disability, behavioral disorders, speech and language difficulties, visual and audiological impairments, cardiac deformities and urogenital problems. Finally, recent research found high prevalence rates (from 10 to 40 times) in children in foster care and in correctional, special education, specialized clinical and Aboriginal populations (95% confidence interval, 4.9–11.7) in global prevalence in the general population [18].

Previous research estimated that 4.3% of children born among heavy drinking pregnant women (defined as an average of two or more drinks per day, or five to six drinks per occasion) will be diagnosed with FAS, which is about three times greater than the quotient estimated among women in the general population who consumed any amount of alcohol during pregnancy [4]. FAS is a preventable disease but its prevalence could increase around the world in the coming years. The averages of alcohol use, binge drinking and drinking during pregnancy are increasing among young women in a lot of countries; moreover, a lot of pregnancies in developing and developed countries are unplanned, with the increased risk of involuntary exposition of the embryo to alcohol in the earliest stage of pregnancy, when brain development is more sensitive to its effects [4].

### 1.3. FASD Prevalence in Adopted Children from Eastern European Countries

The prevalence of FASD among special sub-populations is very high, i.e., in adopted and foster care or orphanage children and particularly in internationally adopted children from countries with a great consumption of alcohol in the general population such as Eastern European countries. FASD can be neurodevelopmentally and behaviorally indistinguishable from other neurodevelopmental disorders. In addition, the lack of expertise in diagnosing FASD and mixed neurodevelopmental and behavioral signs due to abandonment trauma complicate a timely diagnosis [18,19,20,21,22].

According to published data, the prevalence of FASD in children from Russian orphanages is estimated to be between 30% and 66% [7]. A total of 90% of Russian women at fertile age consume alcohol and up to 20% continue to consume it during pregnancy. In a study conducted in Sweden in 2010 with 71 children adopted from Eastern Europe, 52% presented FASD, including 30% FAS, 14% partial FAS and 9% alcohol-related neurodevelopmental disorders [23]. In a cohort of 36 (15 females) adoptees with FASD followed-up for 15.5 years (range 13 to 17), the same authors found that 20 (56%) were dependent on social support. The median intelligence quotient (IQ) of 86 during childhood declined significantly to 71 during adulthood (mean difference: 15.5; 95% CI 9.5–21.4). Psychiatric disorders were diagnosed in 88% of the subjects followed up, especially attention deficit hyperactivity disorder (70%). Three or more disorders were diagnosed in 48%, and 21% had attempted suicide [24,25].

Spain is the second country in the world in the number of international adoptions from Eastern European countries, after the United States. During the period from 2006 to 2018, 23,460 international adoptions took place in Spain, of which 8634 were from Eastern European countries (36.8% of international adoptions). Catalonia is the Spanish Autonomous Community with the most international adoptions in absolute terms, with 5120 adoptions from Russia and Ukraine during the period from 1998 to 2015. [26]

After the clinical observation of a high prevalence of neurocognitive and behavioral disorders among adopted children from Eastern European countries, we hypothesized, as in other countries, that a large number of these adopted children may be also affected by FASD and that FASD could still be underestimated because of numerous undiagnosed and misdiagnosed cases [5,13]. This study aimed to estimate the prevalence of FASD in adopted children from Russia and Ukraine living in Catalonia from a representative sample of adopted children from these European regions.

## 2. Patients and Methods

### 2.1. Sampling Methods and Description

The experimental design was a descriptive observational study of prevalence. Adoptees from Russia and Ukraine were randomly selected from the database of the Catalan Institute for Fostering and Adoption (ICAA), which includes all adoptees from these countries. In total, 3817 children fulfilled inclusion criteria including children from 8 to 24 years of age, adopted from Russia or Ukraine at least 2 years ago and having signed informed consent. The experimental design assumed an expected prevalence of 50% (standard value when the actual prevalence is unknown and which agrees with the prevalence found in other studies) [24], a 90% confidence level, Z = 1.645, an allowable error of 5% and a statistical power of 80% (1β = 0.20). GRANMO free software [27] was used to calculate the estimated sample size with these parameters, obtaining 290 subjects with a maximum loss of 30%.

### 2.2. Recruitment

This study was performed in the context of a cooperation between the Catalan Institute for Fostering and Adoption (ICAA), the Public Health Agency of Catalonia and expert professionals from different hospitals in Barcelona (Hospital Clínic-Maternitat, Hospital San Joan de Déu and Hospital Vall d’Hebron).

A letter explaining the rationale of the study was sent by the Catalan Institute of Foster Care and Adoption (ICAA) to the families of the randomly selected children (existing database) asking them consent to be contacted by clinicians of the collaborating hospitals.

The recruitment process required two rounds of randomization and letter sending until the required sample was reached. Specific software IBM SPSS Statistics for Windows Version 22.0 (IBM Corp; Armonk, NY, USA), was used to randomize twice without removing the cases selected in the previous round. In the first round of submissions, a random sample of 450 children was selected followed by 300 children in the second.

### 2.3. Assessment

Pediatricians and psychologists trained and with experience in FASD clinical diagnostic guidelines performed the assessments using the Institute of Medicine validated diagnostic criteria updated by Hoyme et al. [6] and matched to other validated criteria [7,19,28,29]. The diagnosis was based on a specific clinical assessment and on various cognitive and behavioral standardized neuropsychological tests (Wechsler Intelligence Scale for Children (WISC), Wechsler Adult Intelligence Scale (WAIS), A Developmental NEuroPSYchological Assessment (NEPSY-II), Rey-Osterrieth complex figure, Evaluation of Reading Processes for Secondary Education Students (PROLEC-SE), Child Behavior Checklist (CBCL) and Vineland (VABS-II)), with unified criteria and previous uniform formal training for the professionals from the 3 clinical units [6]. All of the children in the study were evaluated by two different researchers in each group who assessed for signs of dysmorphology associated with FASD, utilizing the lip/philtrum guide of Astley [6], and used a ruler to measure palpebral fissure length. Doubts concerning the diagnosis of some children were discussed by the research group.

For assessment of maternal alcohol consumption, pre-adoption medical records provided by parents and tutors and the ICAA were analyzed in order to find confirmed alcohol consumption during pregnancy and to detect possible neonatal pathologies, as well as other relevant medical or neuropsychological problems. This is a particularly significant problem associated with this patient population because most of the categories of FASD require a confirmed maternal exposure to alcohol.

With regard to neurobehavioral deficits, children underwent a formal developmental assessment: neurocognitive deficits: executive function, intelligence quotient/cognition, learning, memory, visual–spatial; problems with self-regulation: sleep, self-soothing, anger control, attention, impulse control; delayed/deficient adaptive skills: social skills, language, gross/fine motor, daily living skills. Information collected during the evaluation was recorded in a common database.

### 2.4. Statistical Analysis

Statistical analyses were performed by using SPSS v22. Frequencies, means, SDs, SD scores, medians and ranges were calculated for descriptive purposes. The chi-square test and the Fischer test were used to calculate associations between categorical variables with a *p*-value of 0.05. EpiTools (Ausvet; Bruce, Australia) was used to compare proportions and means between our samples with the reference population universe.

### 2.5. Ethical Aspects

The institutional review board (IRB) of each participating hospital approved the study protocol (IRB approval code: 2016/7025/I). Informed consent was obtained from the parents/tutors of all children participating in the study. Families received the diagnostic results in a direct interview (face-to-face, by phone or by teleconference) within 15 days and the assessment report was given by hand or sent encrypted by mail. For queries or follow-up questions, families were told to contact the research unit team or the ICAA office.

## 3. Results

Table 2 details the number of letters sent and the response rate achieved in each round. The rate necessary to get one agreement to participate among the candidate families was one in three. Most letters (63.8%) did not receive a response or were returned by the postal service, mainly due to changes in postal addresses, lack of interest in participating, a long journey to the hospital or because the children were already being assessed somewhere else. However, once contacted, only a small proportion of families (6.53%) refused to participate.

Two hundred and twenty-two (222) families accepted to participate and were randomly allocated to the three research hospitals for assessment: 83 (37.6%) at the Hospital Clínic-Maternitat, 68 (30.6%) at the Hospital Vall d’Hebron and 71 (32.0%) at the Hospital Sant Joan de Déu, all of them in located in Barcelona. Each hospital (through administrative staff or clinical professionals) contacted the corresponding families by phone (preferably) or by email. The purpose of the study was re-explained and any doubts were solved. If one candidate family was not located, the study staff tried to contact them at least three times before discarding.

The mean age of the children recruited was 13.9 years (SD ± 3.31) (minimum of 8 and maximum of 24 years). The mean age at adoption was 2.27 years (SD ± 1.44) (minimum of a few days after birth and maximum of 7 years). Almost two out of three were males (64%) and most, up to 94.1%, were adopted from Russia. A total of 72.5% were adopted using accredited associations. From the 222 participants, 162 cases (73%) were evaluated (Figure 1). There were no significant differences in sociodemographic characteristics between included and not included children.

Of those not evaluated, 18 decided to terminate the study before the clinical evaluation (due to incompatibility of scheduled appointments, living far away or reconsidering their participation due to not wanting to collaborate), 6 had been previously diagnosed with FASD by any of the three teams, 32 were not finally located after the initial contact and 4 did not complete the study.

### General Description of FASD Umbrella Diagnostic Results, Differences by Age, Gender and Year of Adoption

Of the 162 children evaluated, 81 (50%) finally received one of the FASD diagnoses, according to the proposed revised IOM criteria. Of them, 33 (20.4%) had FAS, 28 (17.2%) had partial FAS, 2 (1.2%) had ARBD and 18 (11.1%) had ARND. The FASD diagnoses distribution by gender did not show statistical differences (χ2 = 5.71, df = 4, *p* = 0.222) (Table 3). There were no significant differences in sociodemographic characteristics between participants diagnosed with having FASD or not.

## 4. Discussion

### 4.1. Comparison with Other Studies (Strength of the Study)

The clinical diagnosis performed in children adopted from two Eastern European countries (Russia and Ukraine) showed that 81 children (50%) had FASD, including a diagnosis of FAS in 33 of them (20.4%). This is one of the few studies which have attempted to establish the FASD prevalence rates in the whole population of adopted children from Eastern Europe in a region or country and is the first one in Spain. The percentage of affected children was similar to another study conducted in Sweden in 2010, which obtained a 52% prevalence of affected children in a smaller sample [24], and is lower than the 66% prevalence obtained in orphanages in Russia [23].

Furthermore, adopted children from Eastern European countries are a special subpopulation of people at risk of having FASD [18]. The prevalence of alcohol consumption in the general population is very high, especially between mothers that give up their children for adoption or that lose the safekeeping of them. Usually, they have a very depressed socioeconomic environment where alcohol consumption is still higher. Most of these children are included in international adoption circuits.

It is often difficult to diagnose FASD and the diagnosis may be missed, especially among children who are institutionalized or adopted [19,30]. Reasons for possible underdiagnoses of FASD could be related to lack of awareness and expertise among health professionals regarding the diagnostic features of FASD, absence of systematic data collection, lack of universally accepted diagnostic criteria for the diagnosis of FAS and FASD or unawareness of the importance of listing FASD as a diagnostic entity [10,25].

Another important point to be considered in these adopted children is that deleterious neuropsychological effects due to prenatal exposure to alcohol in adopted children from orphanages are mixed with effects of abandonment trauma [16,22]. In most cases, it is impossible to ascertain what signs depend on alcohol or institutionalization.

Otherwise, in these children, it is very usual to have no validated report of a confirmed consumption of alcohol during pregnancy. As previously mentioned, this point is crucial in FASD diagnosis: Of the 81 children in which fetal alcohol exposure could not be confirmed in our study, 20% of them showed manifestations that would have established a diagnosis of FASD if a history of maternal alcohol consumption was confirmed. Most likely, an important cause of FASD underestimation is due to the information regarding alcohol exposure during pregnancy not being available for many of the adoptive mothers. Among the children with known exposure to alcohol, half met the criteria for a diagnosis of FASD (including FASD, partial FASD, ARBD and ARND).

Another possibility for underdiagnosed FASD is the lack of awareness of healthcare professionals regarding the prevalence of maternal drinking and the diverse manifestations of FASD [19]. Quite often, the history of maternal alcohol consumption is not available to mental health professionals who are evaluating children for developmental or learning disabilities.

It is still important to identify children who have a very high chance of having FASD so that they can receive the appropriate intervention. Diagnosis of FASD at an early stage is beneficial as early intervention may minimize many of the cognitive, behavioral and social problems associated with FASD [19].

### 4.2. Limitations

Despite the efforts made to obtain a representative sample, we cannot completely exclude a selection bias. A large volume of letters did not reach the recipients or were not answered (response rate was 30%). In this context, a possible hypothesis could be the fact that the children did not show any clinically relevant difficulty or, on the contrary, that children were already in clinical follow-up. It may also be due to simpler aspects, such as the original database not being completely updated from the adoption or the fact that families did not understand the meaning of the invitation and the study for some reason, or that they could not participate due to not being able to travel to the visits at hospitals located in Barcelona. However, the tests of comparison between the random sample and the universe of reference showed that they were equal in terms of sex, country of origin and age of adoption but not in the current age, in which the evaluated sample is a little bit younger in age than the reference universe.

### 4.3. Challenges and Opportunities under a Public Health Perspective

Importantly, only 6 out of the 162 children evaluated were previously assessed and diagnosed with FASD, indicating that 75 of the 81 children affected and diagnosed with FASD in this study were never diagnosed before. This implies that expertise in the diagnostic guidelines for FASD is essential for the careful assessment of children with risk factors (prenatally exposed to alcohol, adopted from Eastern European countries, with no classified neuropsychological or behavioral problems, etc.). Countries should invest resources to try to meet the needs of these children by developing diagnostic protocols for health professionals, developing guides for professionals in the fields of education, social and health, designing therapeutic tools and resources for occupational life, daily life and quality of life and adapting and training the child, adolescent and adult mental health resources. The results of this study also show the relevance of introducing improvements in international adoption circuits and protocols [13,19]. Furthermore, we know that the news about FASD in adopted children from these countries associated with changes in legislation is going to produce an important decrease in international adoption.

This study suggests another unknown situation: the prevalence of FASD in native children, not in a sub-population of children at high risk, in order to clarify the real burden of neurocognitive and behavioral problems related to prenatal consumption of alcohol in pregnant women.

## 5. Conclusions

At least half of the children adopted from Eastern Europe (Russia and Ukraine) have a diagnosis of FASD in our study group. Bearing in mind the large physical and functional damage caused by prenatal alcohol exposure, these children and their families might be in need of special mental health treatment and social support. At the same time, it is important to establish an adoption protocol that takes into account this disorder and adequately informs families. As intervention is important and potentially beneficial, it would be imperative to identify children with FASD or at risk for developing FASD at an early stage. There is an urgent need for public attention to be drawn to FASD and for increased efforts to train health professionals on FASD diagnosis, in order to allocate tailored resources to support people affected and their families, to raise awareness about the importance of not drinking during pregnancy and to endorse legislation on the introduction of mandatory labels with health warning messages on alcohol beverage containers.

Only specific and continuing updates for healthcare professionals about drinking habits may have impactful actions to prevent gestational alcohol intake in order to prevent the main cause of mental retardation in Western countries.

## Figures and Tables

**Figure 1 ijerph-18-01388-f001:**
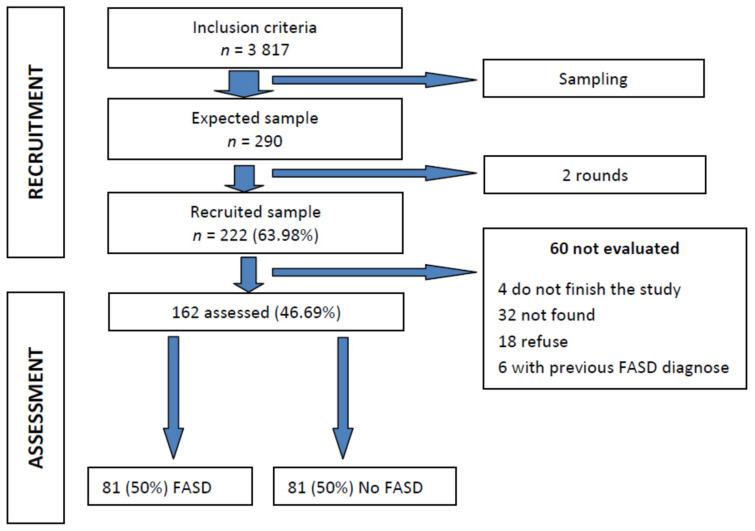
Flowchart of the experimental design and the selection process. Fetal alcohol spectrum disorder (FASD).

**Table 1 ijerph-18-01388-t001:** Diagnostic criteria of fetal alcohol spectrum disorder (FASD) [6].

	Documented Prenatal Alcohol Exposure	Criteria Required	Features
FAS	Yes or no	A to D	A. A characteristic pattern of minor facial anomalies (≥2): 1. Short palpebral fissures (≤10th percentile) 2. Thin vermilion border of the upper lip (rank 4 or 5) 3. Smooth philtrum (rank 4 or 5) B. Prenatal and/or postnatal growth deficiency 1. Height and/or weight ≤10th percentile C. Deficient brain growth, abnormal morphogenesis or abnormal neurophysiology (≥1): 1. Head circumference ≤10th percentile 2. Structural brain anomalies 3. Recurrent nonfebrile seizures D. Neurobehavioral impairment 1. For children ≥3 y of age (a or b): a. With cognitive impairment: −Evidence of global impairment, OR −Cognitive deficit in at least 1 neurobehavioral domain b. With behavioral impairment without cognitive impairment: −Evidence of behavioral deficit in at least 1 domain 2. For children <3 y of age: −Evidence of developmental delay
pFAS	Yes	A and B	A. A characteristic pattern of minor facial anomalies (≥2): 1. Short palpebral fissures (≤10th percentile) 2. Thin vermilion border of the upper lip (rank 4 or 5) 3. Smooth philtrum (rank 4 or 5) B. Neurobehavioral impairment 1. For children ≥3 y of age (a or b): a. With cognitive impairment: −Evidence of global impairment, OR −Cognitive deficit in at least 1 neurobehavioral domain b. With behavioral impairment without cognitive impairment: −Evidence of behavioral deficit in at least 1 domain 2. For children <3 y of age: −Evidence of developmental delay
	No	A to C	A. A characteristic pattern of minor facial anomalies (≥2): 1. Short palpebral fissures (≤10th percentile) 2. Thin vermilion border of the upper lip (rank 4 or 5) 3. Smooth philtrum (rank 4 or 5) B. Growth deficiency or deficient brain growth, abnormal morphogenesis or abnormal neurophysiology 1. Height and/or weight ≤10th percentile, OR 2. Deficient brain growth, abnormal morphogenesis or neurophysiology (≥1): a. Head circumference ≤10th percentile b. Structural brain anomalies c. Recurrent nonfebrile seizures C. Neurobehavioral impairment 1. For children ≥3 y of age (a or b): a. With cognitive impairment: −Evidence of global impairment, OR −Cognitive deficit in at least 1 neurobehavioral domain b. With behavioral impairment without cognitive impairment: −Evidence of behavioral deficit in at least 1 domain 2. For children <3 y of age: −Evidence of developmental delay
ARND	Yes	A and B	A. Documented prenatal alcohol exposure B. Neurobehavioral impairment For children ≥3 y of age (a or b): a. With cognitive impairment: −Evidence of global impairment, OR −Cognitive deficit in at least 2 neurobehavioral domains b. With behavioral impairment without cognitive impairment: −Evidence of behavioral deficit in at least 2 domains
ARBD	Yes	A and B	A. Documented prenatal alcohol exposure B. One or more specific major malformations demonstrated in animal models and human studies to be the result of prenatal alcohol exposure

Note: FAS, fetal alcohol syndrome; pFAS, partial fetal alcohol syndrome; ARND, alcohol-related neurodevelopmental disorder; ARDB, alcohol-related birth defects.

**Table 2 ijerph-18-01388-t002:** Recruitment of the selected adoptees in Catalonia from Russia and Ukraine.

	Letters Sent	Accepted	Refused	Returned by Post	No Response
1st round	450	157 (34.89%)	25 (5.56%)	132 (29.33%)	136 (30.22%)
2nd round	300	65 (21.67%)	24 (8%)	100 (33.33%)	111 (37.00%)
Total	750	222 (29.6%)	49 (6.53%)	232 (30.93%)	247 (32.93%)

**Table 3 ijerph-18-01388-t003:** Distribution by sex of the FASD diagnoses among Russian and Ukrainian adoptees in Catalonia.

			Sex		
			Female	Male	Total
Diagnosis	NO FASD	*n*	34	47	81
		% diagnosis	42.0%	58.0%	100.0%
		% by gender	55.7%	46.5%	50.0%
		% total	21.0%	29.0%	50.0%
	ARBD	*n*	2	0	2
		% diagnosis	100.0%	0.0%	100.0%
		% by gender	3.3%	0.0%	1.2%
		% total	1.2%	0.0%	1.2%
	ARND	*n*	5	13	18
		% diagnosis	27.8%	72.2%	100.0%
		% by gender	8.2%	12.9%	11.1%
		% total	3.1%	8.0%	11.1%
	FAS	*n*	12	21	33
		% diagnosis	36.4%	63,6%	100.0%
		% by gender	19.7%	20.8%	20.4%
		% total	7.4%	13.0%	20.4%
	pFAS	*n*	8	20	28
		% diagnosis	28.6%	71.4%	100.0%
		% by gender	13.1%	19.8%	17.3%
		% total	4.9%	12.3%	17.3%
TOTAL		*n*	61	101	162
		% diagnosis	37.7%	62.3%	100.0%
		% by gender	100.0%	100.0%	100.0%
		% total	37.7%	62.3%	100.0%

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
