# Peer review of "Prevalence of Fetal Alcohol Spectrum Disorders (FASD) among Children Adopted from Eastern European Countries: Russia and Ukraine"

_ijerph, 2021, doi:10.3390/ijerph18041388_

Round 1
Reviewer 1 Report
Dear Author,
With great interest, I have read your paper on the prevalence of FASD among adopted children. This is an important public health topic, especially considering the number of adopted children and the value of early intervention in this group.
Please find below a few comments and questions on your paper:
- Title: I would recommend changing the title to be more specific and in tune with the aim of the study:" This study aimed to estimate the prevalence of FASD in adopted children from Russia and Ukraine living in Catalonia from a representative sample of the adopted children from these European regions"
- Would you consider defining FASD (for example, adding a table with criteria updated Hoyme criteria used in the study)
- Do you consider ND-PAE (neurobehavioral disorder associated with prenatal alcohol exposure) as a part of the FASD spectrum?
- Conclusion: "At least half of the children adopted from Eastern Europe have a diagnosis of FASD" is a bit far stretched and was not proven by the study results. I would suggest naming specific countries as well as adding "in our study group." The high rate of non-responders in the selected study population doesn't give ground for the generalization in the conclusions.
- Would you consider having a control group of adopted children from other geographical regions? Do you have any data on FASD prevalence among adopted children of Spanish origin, for example? It might be interesting to compare, as the adoption process might be connected with social challenges such as maternal alcohol consumption.
- Would you consider adding a passage on adoption trends in Spain. Are international adoptions on the rise? Or due to changes in legislation, do they decrease, as in other countries?
I am looking forward to reviewing the revised version of the paper.
Author Response
- Title: I would recommend changing the title to be more specific and in tune with the aim of the study:" This study aimed to estimate the prevalence of FASD in adopted children from Russia and Ukraine living in Catalonia from a representative sample of the adopted children from these European regions".
The title has been changed.
- Would you consider defining FASD (for example, adding a table with criteria updated Hoyme criteria used in the study).
The Table has been added (Table 1).
- Do you consider ND-PAE (neurobehavioral disorder associated with prenatal alcohol exposure) as a part of the FASD spectrum?
Not exactly, ND-PAE is a category included in DSM-V, but we used Hoyme’s criteria, where ND-PAE is very similar to ARND (Alcohol Related Neurodevelopment Disorder).
- Conclusion: "At least half of the children adopted from Eastern Europe have a diagnosis of FASD" is a bit far stretched and was not proven by the study results. I would suggest naming specific countries as well as adding "in our study group." The high rate of non-responders in the selected study population doesn't give ground for the generalization in the conclusions.
The change has been made.
- Would you consider having a control group of adopted children from other geographical regions? Do you have any data on FASD prevalence among adopted children of Spanish origin, for example? It might be interesting to compare, as the adoption process might be connected with social challenges such as maternal alcohol consumption.
Yes, but there is any data about this point. At now we have a new project in order to ascertain FASD prevalence in children adopted from other countries, including Spanish children, and also in a general population of children at schools.
- Would you consider adding a passage on adoption trends in Spain. Are international adoptions on the rise? Or due to changes in legislation, do they decrease, as in other countries?
We know that international adoptions from these countries are diminishing the last months in part related to news about FASD in adopted children from these countries and changes in legislation.
Reviewer 2 Report
Thank you for the opportunity to review this paper on such an important topic. I hope you find useful my recommendations about this paper.
Title: I think that is very ambitious to use the term “prevalence” for a cohort study in a region of a country. I suggest using another title. Furthermore, the participation rate was low so it is not possible to stablish clearly the prevalence.
Abstract:
-Line 30-45:According to instructions for authors, abstract should include 200 words and the text provided contains more. On the other hand, I suggest removing subheadings such as background, methods, results, etc since recommendations for authors do not include them.
Background.
-Line 59-61: I recommend to support this sentence with a reference.
-Line 63-66: It would be interesting to provide data segregated by gender, if available.
-Line 89-90: In which country? Were there differences among Eastern European countries?.
-Line 92: This study is about mothers from Eastern European countries so I recommend to provide rates for that countries. In this case, Spanish rates are not relevant.
-Line 117: Please, provide the full details for ICD.
-Line 150: Please, provide details for IQ.
I suggest including some idea about the reason why women that give their children up for adoption are high alcohol consumers. It would be interesting to discuss this idea.
Patients and methods.
-Line 175: This estimated prevalence should be supported by a reference. Reference number 24 provided this prevalence.
-Line 197: In order to prove the consistency of measures, please, do provide and cite the scales performed to assess participants. Were children assessed for, at least, two researchers to discuss discrepancies?. Were there doubts with the diagnosis of some children?. Please, check reference number 2 in this paragraph to support the assessment performed to children. That report does not include information about diagnosis.
-Line 210: What kind of measures were performed?. Please, provide details.
-Ethical aspects: Line 219, please provide reference of ethics committee approval.
Where were children assessed?. Was the researcher involved in any other medical procedure to the children? Please, detail setting of the study and circumstances (with/without parents; timing of the evaluation; number of sessions, etc)
Since this research pretends to be a random study, you should to calculate size sample to know if you reached it. Later, you show an expected sample in the flowchart but should be justified in this section.
Results.
-It would be interesting to show a table with sociodemographic characteristics of children that were diagnosed of FASD (etc) and those who are not. According to other studies, children age could be important to consider.
- A table with clinical characteristic of participants diagnosed would be very useful.
-You mentioned that mother´s medical records were reviewed to know the pattern of alcohol consumption during pregnancy. Did you find any result? It would be interesting to show in the results section this data in contrast with children that were diagnosed.
Discussion
-Reasons why the sample size was not reached should be shown as limitations.
Besides, you obtained a loss sample higher than expected. Please, discuss this idea in this section.
-Line 279-81. This idea should be supported by a reference or others studies.
-Line 284-286. This is a result and should be included in results section.
-Limitations: Participation bias should be taken into account. Children with dysfunctional behaviors could be more motivated to participate in the study to know the reason of that problem, so they were diagnosed more often than those children that did not show any features through the years.
-It would be interesting to discuss about the diagnosis of FASD in different ages as other studies suggest.
-My recommendation is to extend discussion section with the main finding of the study, rate of FASD in adopted children, in contrast with sociodemographic characteristics and maternal consumption of alcohol.
Author Response
Title: I think that is very ambitious to use the term “prevalence” for a cohort study in a region of a country. I suggest using another title. Furthermore, the participation rate was low so it is not possible to establish clearly the prevalence.
We changed the title of the manuscript and added some more explanation to the participation rate. Out of the data included in the manuscript we analyzed the medical records of selected but not included children and there were no differences with the included ones about sociodemographic nor clinical data from adoption date.
Abstract
-Line 30-45: According to instructions for authors, abstract should include 200 words and the text provided contains more. On the other hand, I suggest removing subheadings such as background, methods, results, etc since recommendations for authors do not include them.
Abstract has been reduced and subheadings have been removed.
Background
-Line 59-61: I recommend to support this sentence with a reference.
A reference has been added.
-Line 63-66: It would be interesting to provide data segregated by gender, if available.
There were no significative differences by gender (χ2 = 5,71, df = 4, p = 0,222).
-Line 89-90: In which country? Were there differences among Eastern European countries?
This data come from a global study (ref. 12) of prevalence of FASD among children and youth, including not only Eastern European countries. We studied Russia and Ukraine because most of adopted children from East Europe in our geographical area come from these 2 countries. It has been published prevalence of FASD in Russia (36.5%) (refs. 9 and 12) and in orphanages in Russia (66%) (ref. 23).
-Line 92: This study is about mothers from Eastern European countries so I recommend to provide rates for that countries. In this case, Spanish rates are not relevant.
Data from Eastern European countries are included in the next paragraphs. Data from Spain are important in order to compare prevalence rates between Spain and countries of origin of adopted children.
-Line 117: Please, provide the full details for ICD.
Details have been included.
-Line 150: Please, provide details for IQ.
Details have been included.
I suggest including some idea about the reason why women that give their children up for adoption are high alcohol consumers. It would be interesting to discuss this idea.
This point has been included in the manuscript.
Patients and methods
-Line 175: This estimated prevalence should be supported by a reference. Reference number 24 provided this prevalence.
The reference has been included.
-Line 197: In order to prove the consistency of measures, please, do provide and cite the scales performed to assess participants. Were children assessed for, at least, two researchers to discuss discrepancies?. Were there doubts with the diagnosis of some children?. Please, check reference number 2 in this paragraph to support the assessment performed to children. That report does not include information about diagnosis.
The scales were the recommended by WHO: Wechsler (WISC, WAIS), NEPSY-II, Rey-Osterrieth complex figure, PROLEC-SE, CBCL, Vineland (VABS-II).
Previously to the beginning of the study, all the clinical researchers standardized the clinical assessment. During the study, physical exam was made at any time by 2 different researchers.
Doubts with the diagnosis of some children were discussed by the research group.
The reference has been deleted.
-Line 210: What kind of measures were performed?. Please, provide details.
A comment has been included.
-Ethical aspects: Line 219, please provide reference of ethics committee approval.
The referebce has been added.
Where were children assessed?. Was the researcher involved in any other medical procedure to the children? Please, detail setting of the study and circumstances (with/without parents; timing of the evaluation; number of sessions, etc.).
The children were assessed at each hospital participating in the project.
No, the researcher was not involved in any other medical procedure to the children.
The assessment was made with parents (physical exam) and without them (neuropsychological assessment), in the morning and, in a sole session.
Since this research pretends to be a random study, you should to calculate size sample to know if you reached it. Later, you show an expected sample in the flowchart but should be justified in this section.
Thanks for the comment. Sample size was calculated using GRANMO free software with the following parameters: expected prevalence of 50% (standard value when the actual prevalence is unknown, and which agrees with the prevalence found in other studies), 90% confidence level, Z=1,645, allowable error 5% and statistical power of 80% (1-β= 0,2).
The estimated sample size was 290 subjects with a maximum loss of 30%.
The parameters used in the sample size calculation and the sample size obtained have been described in detail and included in “Patients and Methods. Sampling methods and description” following reviewer suggestion.
Flowchart has been modified.
Results
-It would be interesting to show a table with sociodemographic characteristics of children that were diagnosed of FASD (etc.) and those who are not. According to other studies, children age could be important to consider.
A comment has been included in the manuscript.
Children age data is also considered in the manuscript.
- A table with clinical characteristic of participants diagnosed would be very useful.
A comment has been included in the manuscript.
-You mentioned that mother´s medical records were reviewed to know the pattern of alcohol consumption during pregnancy. Did you find any result? It would be interesting to show in the results section this data in contrast with children that were diagnosed.
Medical records of the mother, specifically related to alcohol consumption during pregnancy, were reviewed, but only when data were available.
A comment about this point is included in the discussion.
Discussion
-Reasons why the sample size was not reached should be shown as limitations.
This comment is included in Limitations section.
Besides, you obtained a loss sample higher than expected. Please, discuss this idea in this section.
This point is included in the manuscript.
-Line 279-81. This idea should be supported by a reference or others studies.
A reference has been added.
-Line 284-286. This is a result and should be included in results section.
The text has been modified.
-Limitations: Participation bias should be taken into account. Children with dysfunctional behaviors could be more motivated to participate in the study to know the reason of that problem, so they were diagnosed more often than those children that did not show any features through the years.
We have discussed this point. Sociodemographic characteristics and also health problems (from medical records) were very similar in both groups of included and not included children.
-It would be interesting to discuss about the diagnosis of FASD in different ages as other studies suggest.
We included a limited range of ages in order to include children arrived at least 2 years ago and with a range of age from 8 to 24 years. The diagnostic criteria of FASD are the same at every age (except for children under 3 years old). This point could be very interesting in general population.
-My recommendation is to extend discussion section with the main finding of the study, rate of FASD in adopted children, in contrast with sociodemographic characteristics and maternal consumption of alcohol.
Thank you for your suggestion. We have extended the discussion in the manuscript.
Reviewer 3 Report
Overall an important study with a few minor typos - I list these below.
Minor Corrections, Colom et al, 2021.
Page 2 keywords – use of mental retardation – many people with FASD and indeed with no FASD find this term outdated and offensive, please find another word (s) and replace this term throughout the article – possibly use the words neuro-cognitive or neuro-developmental impairments.
Page 2 lines 86 to 94
The para as a whole is focused on alcohol use during pregnancy – but on line 92, FASD prevalence for only one country, Spain, is introduced – this fact should be moved to any of the later paragraphs on page 3. The end of this paragraph, lines 93 and 94, therefore also need reworking to conclude the focus on alcohol use during pregnancy rather than FASD, because the later paras focus on FASD prevalence, not this one.
Page 3 line 99-101 – the 25.2% alcohol use during pregnancy fact is repeated here from page 2 line 91 and 92 so one of them should be deleted.
Page 3 para two and line 106/107 the UK study is a SCREENING prevalence study and this should be clarified.
Page 3 para 3 and line 109 -110, we have a third repeat of the 25% alcohol use during pregnancy of the general European population fact; hence delete or reword.
Page 3 para 4 line 123-124 – this is a repeat of para two line 113 but if you delete the words ‘compared with the 7.7% per 1,000’ to the end of the sentence this will be fine.
Line 141 add the word ‘trauma’ alongside abandonment.
Page 5 line 208 this line does not make sense, needs tense work – children underwent or had a formal developmental assessment is suggested.
Discussion
Page 7 line 280 add trauma alongside abandonment.
Page 7 line 282 delete ‘missed’ and add ‘to have no’ validated report.
Author Response
Page 2 keywords – use of mental retardation – many people with FASD and indeed with no FASD find this term outdated and offensive, please find another word (s) and replace this term throughout the article – possibly use the words neuro-cognitive or neuro-developmental impairments.
Keywords have been changed.
Page 2 lines 86 to 94
The para as a whole is focused on alcohol use during pregnancy – but on line 92, FASD prevalence for only one country, Spain, is introduced – this fact should be moved to any of the later paragraphs on page 3. The end of this paragraph, lines 93 and 94, therefore also need reworking to conclude the focus on alcohol use during pregnancy rather than FASD, because the later paras focus on FASD prevalence, not this one.
The text has been corrected.
Page 3 line 99-101 – the 25.2% alcohol use during pregnancy fact is repeated here from page 2 line 91 and 92 so one of them should be deleted.
The text has been deleted.
Page 3 para two and line 106/107 the UK study is a SCREENING prevalence study and this should be clarified.
The text has been clarified.
Page 3 para 3 and line 109 -110, we have a third repeatvof the 25% alcohol use during pregnancy of the general European population fact; hence delete or reword.
The text has been deleted.
Page 3 para 4 line 123-124 – this is a repeat of para two line 113 but if you delete the words ‘compared with the 7.7% per 1,000’ to the end of the sentence this will be fine.
The text has been deleted.
Line 141 add the word ‘trauma’ alongside abandonment.
The word has been added.
Page 5 line 208 this line does not make sense, needs tense work – children underwent or had a formal developmental assessment is suggested.
The text has been changed.
Discussion
Page 7 line 280 add trauma alongside abandonment.
The word has been added.
Page 7 line 282 delete ‘missed’ and add ‘to have no’ validated report.
The text has been changed.
Round 2
Reviewer 2 Report
Some of the amendments were covered partially. However, the manuscript was improved substantially.